# Foliar Application of Silicon in *Vitis vinifera*: Targeted Metabolomics Analysis as a Tool to Investigate the Chemical Variations in Berries of Four Grapevine Cultivars

**DOI:** 10.3390/plants11212998

**Published:** 2022-11-07

**Authors:** Stefania Sut, Mario Malagoli, Stefano Dall’Acqua

**Affiliations:** 1Department of Pharmaceutical and Pharmacological Sciences, University of Padova, Via Marzolo 5, 35121 Padova, Italy; 2Department of Agronomy, Animals, Food, Natural Resources and Environment, University of Padova, Viale dell’Università 16, 35020 Padova, Italy

**Keywords:** anthocyanins, amino acids, glucose, LC-DAD-MS, phenolics

## Abstract

Silicon (Si) is a beneficial element for the growth of various crops, but its effect on plant metabolism is still not completely elucidated. Even if Si is not classified as an essential element for plants, the literature has reported its beneficial effects in a variety of species. In this work, the influence of Si foliar application on berry composition was evaluated on four grapevine cultivars. The berries of Teroldego and Oseleta (red grapes) and Garganega and Chardonnay (white grapes) were analyzed after foliar application of silicon by comparing the treated and control groups. A targeted metabolomic approach was used that focused on secondary metabolites, amino acids, sugars, and tartaric acid. Measurements were performed using liquid chromatography coupled with a diode array detector and mass spectrometry (LC-DAD-MS^n^), a LC-evaporative light scattering detector (ELDS), and LC-MS/MS methods specific for the analysis of each class of constituents. After the data collection, multivariate models, PCA, PLS-DA, OPLS-DA, were elaborated to evaluate the effect of Si application in the treated vs. control samples. Results were different for each grape cultivar. A significant increase in anthocyanins was observed in the Oseleta cultivar, with 0.48 mg g^−1^ FW in the untreated samples vs. 1.25 mg g^−1^ FW in the Si-treated samples. In Garganega, Si treatment was correlated with increased proline levels. In Chardonnay, the Si application was related to decreased tartaric acid. The results of this work show for the first time that Si induces cultivar specific changes in the berry composition in plants cultivated without an evident abiotic or biotic stress.

## 1. Introduction

Silicon (Si) has a controversial role in plant grow, it is not considered an essential element, but the role of Si is complex and it is involved in a variety of mechanisms in regulating nutrient deficiency and toxicity in different plant species [1]. It is widely described that Si can be a beneficial element for the growth of various crops, limiting the effects of biotic and abiotic stresses in plants and alleviating the effects of plant diseases [2,3,4,5,6]. The literature has reported the beneficial roles of silicon on plant growth in adverse environmental conditions [2,5,6,7]. The established Si-induced mechanisms to improve the plants’ resistance to biotic and abiotic stresses take place in the soil, in the root system, and inside the plant [3]. Si can be applied to soil, and several studies have shown its role on nutrient availability on the rhizosphere and root uptake [1]. On the other hand, this element can also be used by foliar application, as recently described for bean plants where Si foliar application has been used to attenuate potassium deficiency [8].

As described in a review, the foliar application of silicon has a positive effect under stressful conditions for plants such as salinity, deficiency or excess of water, high and low temperature, and strong pressure of diseases and pests [9]. The first proposed hypothesis to explain how silicon foliar application reduces or impedes fungal penetration was the formation of a mechanical barrier below the leaf cuticle and in the cell wall due to silicon polymerization [10]. However, new insights suggest that silicon effects on plant resistance may also occur through mediated host plant resistance mechanisms against pathogen infection [10]. As a further general explanation of the improved tolerance to pathogens in plants supplied with Si, the literature suggests the activation of the phenylpropanoid pathway, resulting in increased total soluble phenolics and lignin [10].

Several examples of studies considered the application of Si in viticulture, showing its usefulness [11,12,13,14,15]. The tolerance against abiotic and biotic stress in plants of *Vitis vinifera* L. cv. Grüner Veltliner was enhanced by foliar and soil silicon applications [11]. The authors reported that the soil application of silicon increased plant-available Si, but only foliar application increased the total silicon concentrations in the leaves, the fruit yield, and the grape cluster weight. The authors provided evidence for the potential, or at least partial, replacement of conventional fungicides, with the consequence of a more sustainable viticulture in terms of soil protection and biodiversity [11].

Another paper reported the chemical composition of red wines obtained after the application of monosilicic acid to grapevines grown in a viticultural area where fungal diseases during summer are important [12]. The results revealed that wine obtained from the Si-treated grapes presented a higher content of total phenols, total anthocyanins, total tannins, and lower levels of gluconic acid and glycerin, two common compounds of botrytized grapes [12]. Up to now, relatively few studies have considered the effects of the silicon foliar application on the composition of grape berries, while most of the studies focused on the chemical constituents of wine and its quality characteristics. For this reason, in this paper, we decided to consider Si foliar application on grapevines and to study the berry composition.

Establishing the possible relationships between Si application and berry metabolites can be a challenge due to the complex phytoconstituents and the multiple metabolic pathways involved [11,12,13]. In this regard, new information can be obtained by applying a metabolomic based approach. Metabolomics deals with the study of low molecular weight compounds in biological tissues and can be ideal to evaluate the effects of treatments or external stimuli on plants [16]. Untargeted or targeted approaches can be used to discover new compounds involved with the considered treatment or to evaluate quantitative changes in the selected known metabolites, respectively. Such approaches have been used, for example, for the analysis of wines obtained from organic biodynamic and conventional grapes [17,18] or to evaluate modifications in grapevines obtained under conventional or biodynamic management [19]. In a previous work, we studied the grapevine Garganega cultivar treated with Si-based preparation in biodynamic management. Leaf and berry samples subjected to untargeted ultra-pressure liquid chromatography-quadruple time of flight (UPLC-QTOF) metabolomics analysis revealed increased synthesis of shikimate related secondary metabolites [20], showing a relationship between Si foliar application and the biosynthesis of secondary metabolites.

The literature indicates that Si can influence the plant nutrient uptake and plant secondary metabolites [1,21]. Up to now, studies related to Si application in *Vitis vinifera* have mainly considered plants subjected to biotic or abiotic stress such in the semiarid climate in Brazil [14], in the presence of botrytis infection [12], or grapes grown in calcareous grey desert soil [15].

In this paper, we studied the effects of Si foliar application in *Vitis vinifera* plants to assess whether the treatment, without evident biotic or abiotic stress, can influence the berry composition. For this reason, amino acids, tartaric acid, and sugar levels as well as phenolics have been measured in berries. To establish the possible effects of silicon foliar application on the chemical composition of the berries, a targeted metabolomic was used aimed to establish whether the treatment can affect the primary and secondary metabolites of berries. To assess whether a common effect of Si-treatment could be evidenced in the berries of the four grape cultivars, two red (Teroldego and Oseleta) and two white (Garganega and Chardonnay) were investigated. Amino acids, sugars, and tartaric acid were chosen because these compounds are important for berry quality. Secondary metabolite contents, in particular polyphenols, were measured due to their presence in the berries and due to the reported influence of Si on their biosynthesis. Thus, the evaluation of these metabolites can help to depict the possible effects of Si on *Vitis vinifera*. The target metabolomic approach merged data obtained with liquid chromatography tandem mass spectrometry (LC-MS/MS) for the amino acid analysis, liquid chromatography evaporative light scattering detector (LC-ELSD) for sugar and tartaric acid determination, and liquid chromatography diode array multiple stage mass spectrometry (LC-DAD-MS^n^) for the analysis of polyphenols. Furthermore, the selected metabolites were chosen due to their possible interest in winemaking (sugars, acids) and for the specific properties of the wines (anthocyanins and phenolics). Due to the complex nature of Si interaction with plants, the target metabolomic approach describing metabolic changes could offer new information to better explain the effects induced by the foliar application of silicon in *Vitis vinifera*.

## 2. Results

To assess possible variations induced by Si treatment, different analyses were performed on the berries. First, carbon–nitrogen–sulfur (CNS) measurements were performed and no significant changes were observed in those elements in the berries of the four considered cultivars (data are summarized in Appendix A).

### 2.1. Secondary Metabolite Fingerprinting of the Berries of the Four Grape Cultivars

The phytochemical fingerprints of berry secondary metabolites for the four cultivars were obtained. The LC-DAD-MS^n^ method was used to acquire the data in both positive and negative ion mode to obtain information on anthocyanins and other phenolics, respectively. UV–Vis spectra of the peaks, obtained by DAD, are helpful for the identification of the compound classes. Furthermore, the DAD data were used to obtain quantitative data for each identified compound. The fingerprint varied with the cultivar, as shown in the chromatograms reported in Appendix A. Each chromatogram was investigated by merging the information obtained by the DAD detector (UV spectra) and MS detector (*m/z* value and fragmentation spectra) to obtain an accurate identification of the secondary metabolites for each cultivar. Where possible, reference compounds were used to confirm the identity of the constituents and specific compounds were used to quantify the different compounds grouped in three classes of phytoconstituents, namely, catechin for procyanidin, catechins, rutin for flavonoids, cyanidin for anthocyanin, and the related derivatives.

### 2.2. Quali-Quantitative Fingerprint of Secondary Metabolites in Berries

A similar composition was observed for the two red cultivars Oseleta and Teroldego, with anthocyanins and procyanidins (PAC) being the main compounds. An enlarged portion of the chromatogram is reported in Appendix A. Anthocyanin glucosides, acetyl glucosides, and *p*-coumaroyl glucosides were the three main categories identified. Malvidin-3-O-glucoside and delphinidin-3-O-glucoside while malvidin-3-O-glucoside and peonidin-3-O-glucoside were the most abundant compounds in the Oseleta and Teroldego berries, respectively. The total amount of anthocyanin was higher in Teroldego (1.3 mg g^−1^) than in Oseleta (0.85 mg g^−1^) (Table 1). In the analysis of PAC, catechin and PAC tetramer were the most abundant compounds (Table 1). The total amount of PAC was double in Teroldego (0.25 mg g^−1^) compared to Oseleta (0.12mg g^−1^).

A similar composition was observed for the two white cultivars Garganega and Chardonnay, with flavonoid glycosides and procyanidins (PAC) as the main compounds (Appendix A). The flavonoid amount was higher in Garganega (0.057 mg g^−1^) than in Chardonnay (0.011 mg g^−1^), while PAC was higher in Chardonnay (0.177 mg g^−1^) than in Garganega (0.138 mg g^−1^). Quercetin-3-O-glucoside was the most abundant compound in Garganega while epicatechin and catechin accumulated more in Chardonnay. The quali-quantitative data related to the berries of the four grape cultivars are summarized in Table 1. Anthocyanins were not detectable in the white berries (Garganega and Chardonnay) while flavonoid glycosides were not detectable in the red berries (Terodego and Oseleta).

### 2.3. Multivariate Analysis

Given the characteristic berry fingerprint each grape cultivar displays, we decided to perform a metabolomic target analysis with the aim to compare the four grape compositions after Si application. Thus, the grouping of samples will be described based on the variations in the amounts of secondary metabolites, tartaric acid, sugars, and amino acids.

At first, the PCA elaboration of the whole dataset of berries showed aa grouping of red and white grapes (Figure 1A) with PC1 and PC2 explaining a total of 55.9% of the observed variance. The classes of secondary metabolites discriminating the two groups of berries (white and red) can be observed in the score plot (Appendix A). As expected, the anthocyanidin derivatives were correlated with red berry group, while other polyphenols were related with white grapes, mainly Garganega. Glucose, fructose, and tartaric acid were discriminant for the red grapes. Amino acids were associated with the white berries, mostly Chardonnay.

The absence of grouping was observed when comparing the Si-treated and control berries of the four grape cultivars, suggesting a non-detectable common effect due to the silicon treatment (Figure 1B).

To more deeply evaluate the potential influence of Si treatment on the berry composition of the four grape cultivars, the dataset for each cultivar was separately elaborated. Figure 1C reports the PCA for the treated and control berries of the four grape cultivars. PC1 and PC2 explained a total of 73.8% of the observed variance for the Oseleta grapes, 65% for Teroldego, 58.7 for Chardonnay, and 66.2 for Garganega. Only the PCA of the Oseleta berries showed grouping of the treated and control samples.

In Oseleta, the amino acids and anthocyanidins were the classes of compounds pinpointing the differences between the treated and control berries (Appendix A). Valine, asparagine, phenylalanine, arginine, and tryptophan contents decreased (*p* < 0.05 T vs. C), while the total anthocyanidin and tartaric acid content significantly increased (*p* < 0.05 T vs. C) in the treated vs. control samples. The amount of total procyanidins, glucose, and fructose increased in the treated berries and the total amino acids decreased, although not significantly (Table 2).

The results obtained by the non-supervised method indicate that Si foliar treatment induced different response in the four considered grape cultivars.

Supervised multivariate elaboration can help to highlight differences among groups and to obtain more information from the dataset. The partial last square discriminant analysis (PLS-DA) was conducted with the aim to highlight fine changes ascribable to the silicon treatment. In PLS-DA, evident grouping was obtained for all of the cultivars (Figure 2). The data confirmed the results obtained for Oseleta berries with the PCA, showing that the most discriminant compounds were the anthocyanins, whose amount increased in the treated samples.

The score plot evidenced that the descriptive variables of the Teroldego treated berries were asparagine, a procyanidin trimer isomer 2, malvidin-3-O-(6-*p*-coumaroyl)glucoside, peonidin-3-O-(6-*p*-coumaroyl)glucoside, and delphinidin-3-O-(6″-acetyl-glucoside) (Appendix A). Fructose, glucose, procyanidin trimer isomer 1, and cyanidin-3-O-(6″-acetyl-glucoside) decreased in the treated vs. control berries. The results are summarized in Table 3 and show that the sugars decreased in the treated sample, but the changes were only statistically significant for fructose.

Histidine, tyrosine, serine, and leucine were the descriptive variables of the treated Chardonnay berries, according to the score plot (Appendix A). Procyanidin trimer isomer 2, catechin, and epicatechin decreased in the treated samples, in agreement with the total PAC content. Tartaric acid decreased in the treated berry (*p* < 0.05), while no effect on the contents of glucose and fructose was evidenced due to the silicon application (Table 2).

Garganega treated berries showed an increment of proline (Appendix A), procyanidin trimer isomer 2, procyanidin tetramer, and catechin gallate, as summarized in Table 3. The content of glucose, fructose, total Pac, flavonoids, and tartaric acid did not significantly change when comparing the treated vs. control berries. The total level of amino acids increased in the treated samples, but not significantly (Table 2).

Once a qualitative indication by the multivariate analysis was obtained, the quantitative changes in the selected metabolites in the berries of each cultivar were estimated (Table 3). All of the metabolites identified as main descriptors for Oseleta were statistically different in the treated compared to the control samples. Fructose, cyanidin-3-(6″ acetyl) glucoside in the Teroldego and proline in the Garganega treated berries were significantly changed, while no significant changes were recorded in Chardonnay (Table 3). The decrease in the amount of sugars and the increase in polyphenols, in particular two anthocyanins and one procyanidin derivative, were observed in the Teroldego treated berries. In the Garganega berries, the application of silicon induced the decrease in arginine and the increase in proline (*p* < 0.05) and some polyphenols. In contrast, the level of some polyphenols decreased in the silicon-treated Chardonnay berries, and the amount of histidine, tyrosine, serine, and leucine increased, although the level of total amino acids did not vary.

The separate elaboration for each cultivar revealed the lack of a common trend in the changes in the targeted metabolites consequent to silicon foliar application, thus suggesting that the foliar application of Si stimulated cultivar-specific metabolic changes in the berries.

A further data elaboration was performed, merging the data of the two red cultivars from one side and the two white cultivars from the other, with the aim to uncover possible common metabolic responses to the silicon supplementation related to the color of the berries. The datasets were elaborated using the supervised approach, namely orthogonal partial last square discriminant analysis (OPLS-DA), in which the differences between the control and treated berries are represented on the horizontal axis, while the differences among the cultivars are mostly described on the vertical axis. In the OPLS-DA of red berries (Figure 3A) and white berries (Figure 3B), partial clusterizations were observed among the control and treated samples in the x axes of the plot. Comparing the two graphs, the clusterization between the T vs. C samples was more emphasized in red berries.

Apparently, the silicon foliar application stimulated the anthocyanin production in the red grapes. In the score plot, the anthocyanins were all in the +x part (Appendix A). Delphinidin-3-O-glucoside (*m/z* 465), peonidine-3-O-glucoside (*m/z* 479), and malvidin-3-O-glucoside (*m/z* 493) were the descriptive variables of the treated red berries, according to the score plot.

A less evident discrimination resulted in the loading plot of white berries (Figure 3B) compared to red berries. In the loading plot of the white grapes (Appendix A), most of the descriptors were close to the *y*-axis, suggesting the presence of a moderate correlation between the control and treated group. The changes ascribed to the silicon treatment were related to proline and some procyanidin derivatives in the white cultivars. To summarize, the foliar application of silicon induced anthocyanin synthesis in red grapes and mainly promoted proline accumulation in white grapes.

Finally, all of the data were merged, and the OPLS-DA allowed us to discriminate the control vs. treatment in the *x*-axis while the red–white berries were separated in the *y*-axis (Figure 3C). The compounds that discriminated the red versus white berries were, as expected, the anthocyanins, but significant variations in the amino acid composition were also observed such as serine, alanine, aspartic acid, and histidine (Appendix A). On the other hand, the metabolites that were correlated with the differences in the C vs. T samples were less descriptive, as the dots representing the variables all populated the graph close to the central vertical axes. No strong correlation was observed with treatment, and descriptors in the +x part of the graph were also close to the central axis. Proline and catechin-3-gallate (*m/z* 441) and procyanidin tetramer (*m/z* 1153) were the metabolites more correlated with the silicon treatment (Appendix A).

## 3. Discussion

The berries of four grape cultivars, Teroldego and Oseleta (red grapes), Garganega and Chardonnay (white grapes), were used as the model to observe the metabolic changes of berries induced by Si foliar application. The multivariate analysis approach allowed us to depict the changes. Samples were grouped first for each cultivar, then for red and white grapes, and finally in whole samples of the four cultivars to assess whether a common trend was detectable.

In red wine, polyphenols are natural key compounds responsible for quality, sensory characteristics (astringency, bitterness, color), and aptitude for aging. Thus, the increment of anthocyanins in berries can be valuable in terms of the berry composition for the final quality of the wine.

Previous works related to Si foliar application evaluated the variations in grape composition under biodynamic management in which silicon was applied. Parpinello et al. (2019) reported that the contents of phenolic compounds significantly decreased in wines after the first year of treatment with Si preparation, while in the second year, polyphenols increased in biodynamic wine [22]. A recent study reported that the total anthocyanin content in mature berries of the Touriga Nacional and Touriga Franca grapevine varieties increased, although not significantly, in plants treated with potassium silicate compared to the control [23]. Particularly, the berry skins of the plants subjected to silicon treatment showed the highest concentration of glycosylated anthocyanins. Malvidin-3-glucoside was observed as the predominant anthocyanin, and in general, glycosylated anthocyanins marked an increase in concentration in the silicon-treated samples [23]. In agreement with these findings, we found that after foliar Si-application, the berries of the two red cultivars Teroldego and Oseleta presented increased levels of mono glycosylated anthocyanins, namely, delphinidin-3-O-glucoside, peonidin-3-O-glucoside, and malvidin-3-O-glucoside. Glycosylation is an important modification to increase the hydrophilicity and stability of anthocyanins, as the anthocyanidins are inherently unstable under the physiological conditions. In *V. vinifera* berries, anthocyanidins can only be O-glycosylated at the C3 position with the addition of glucoses by the activity of UDP-glucose: anthocyanidin: flavonoid glucosyltransferase [24]. Anthocyanins are synthesized by an extremely complex network of all of the structural enzymes in the pathway [24]. It is speculated that for the efficient biosynthesis of anthocyanins, all the key enzymes involved are associated with each other to form a multi-enzyme complex [25,26,27]. However, the working mechanism of the entire complex in vivo still needs to be clarified, particularly for grapes. It is known that the color variation of the grape berries corresponds to the pattern of genotype-specific expression of the whole set of genes involved in anthocyanin biosynthesis, in a direct transcript–metabolite–phenotype relationship [24]. In another work, silica nanoparticles were used on vitis plants by foliar application and the authors reported that the phenolic content and ascorbic acid was increased in berries. Furthermore, the authors claimed that the treatment induced photosynthetic pigments [28].

Our findings are in agreement with a recent publication that showed increased anthocyanin contents in wines obtained from Si-treated grape plants [12]. Our observation that the Si foliar application increased anthocyanin in grape berries suggests a possible role of this element in the complex biochemical pathway of these secondary metabolites.

No major differences were observed in the berry metabolites of the two white grape cultivars treated with Si. Glucose and fructose decreased slightly and the total amino acid content increased. In Garganega berries, the amino acid proline significantly increased after Si application. Accumulation of proline is believed to have adaptive roles, enhancing tolerance to several plant stresses such as salt, drought, low and high temperature, heavy metal, anaerobiosis, UV irradiation, atmospheric pollution, nutrient deficiency, and pathogen infection [29]. Proline can act as an osmoprotective and protect cell components from oxidative stress. Accumulation of free proline is likely to exert a protective action on grape cellular processes, and the level is usually enhanced in the late stages of ripening, coinciding with an increasing concentration of sugars, which leads to the intensification of osmotic pressure [30]. However, the recorded rise in proline content in Garganega berries was not linked to a higher sugar concentration.

The increase in some polyphenols in the white grapes, namely, catechin-3-gallate (*m/z* 441) and procyanidin tetramer (*m/z* 1153), suggests a role of Si foliar application in influencing the phenolic composition of the berries, with possible positive effects on the quality of the musts and wines. Proanthocyanidins are responsible for some major wine sensorial properties (astringency, browning, and turbidity) and are involved in the wine aging processes. Specific studies have focused on grape seed, and skin proanthocyanidins indicate direct links to significant changes in the developing berry [31]. The influence of silicon treatment on secondary metabolite synthesis was recently evidenced, with the increased level of catechin, epicatechin, and other phenolics in must obtained from Chardonnay berries treated with sodium metasilicate [13]. In a previous paper, we described the stimulation of the biosynthetic pathways of phenolics in the leaves and berries of Garganega cultivar plants treated with a Si-based preparation, with a contrasting response due to the site of the vineyard [20]. A recent study on buckwheat plants subjected to Al^3+^ toxicity demonstrated the influence of Si treatment on the phenylpropanoid pathway [32].

The overall results obtained by the multivariate approach indicate that the treatment with Si of the four different grape cultivars without evident abiotic or biotic stress induced significant changes in some cultivar specific metabolites. Further studies will be focused on the same cultivar response to Si in the presence of biotic and abiotic stress to observe whether the modified metabolites will be the same.

## 4. Materials and Methods

### 4.1. Experimental Sites and Plant Samples

The experiment was conducted with the participation of four viticultural farms located in North Italy. In each farm, a grape cultivar, representative of the viticultural “terroir”, was selected. The red grape Oseleta (Os) in the Verona region, the red grape Teroldego (Te) in the Trento region, the white grape Garganega (Ga) in the Vicenza region, and the white grape Chardonnay (Ch) in the Bologna region. In each farm, two plots (C—control and T—Si-treated) of 20 plants each were identified in the vineyard. Between the two plots, two rows of untreated plants were left as a buffer. In the T plots, plants were spread with a solution of 3g SiO_2_ powder dissolved in 50 L of water and mixed for 1 h. The volume is sufficient to treat a 1 ha canopy. The solution was manually spread over the grape canopy in the early morning on 10 May for Ch, 16 May for Ga, 21 May for Os, and 24 May for Te, 2019. Concomitantly, the control plants were only sprayed with the same volume of water.

At harvest time (17 September for Ga, 19 September for Ch, 4 October for Os, and 10 October for Te), about 100 berries per plant were collected from 12 tagged plants in both the C and T plots, placed in plastic bags, immediately frozen, taken to the lab, and stored at −20 °C for later analyses. Prior the analyses, the samples of each cultivar were mixed, keeping separate the control from the treated berries. Each biological replicate was obtained from three pool samples and analyzed in triplicate.

### 4.2. CNS Measurement

The berries were dried completely in an oven at 80 °C for 48 h and then crushed to make a fine powder. Ten mg of powdered samples were used to measure the carbon, nitrogen, and sulfur content by a vario micro cube instrument run in a CHNS analyzer (Elementar Vario ELIII, Elementar, Langenselbold, Germany).

### 4.3. Sample Preparation for Secondary Metabolite Analysis

There were eight groups of samples (control and Si-treated for each grape cultivar). Six specimens of 30 g were collected from each of the eight groups. About 30 g of berries from each cultivar were rapidly thawed to room temperature, and 15 g was rapidly homogenized in a mixer. Two g of the homogenized material was added with 4 mL of a mixture 50% (*v*/*v*) of methanol/water containing 5% of formic acid. The obtained samples were sonicated for 10 min and centrifugated at 13,000 rpm for 10 min. The supernatant was filtered at 0.45 µ and used for LC analysis and the quantification of secondary metabolites, sugars, tartaric acid, and amino acids.

### 4.4. Secondary Metabolite Analysis

An Agilent 1260 chromatograph (Santa Clara, CA, USA) equipped with a 1260 diode array detector (DAD) and Varian MS-500 ion trap mass spectrometer equipped with an ESI source were used. At the end of the column, two “T” splitters separated the flow rate: half of the liquid was split to DAD and half to the Agilent/Varian MS-500 ion trap mass spectrometer (Palo Alto, California). UV–Vis spectra were acquired in the range of 190–600 nm. The LC separation was obtained on an Agilent C-18 XDB (Santa Clara, United States) 3.0 × 150 mm (3 micron), as mobile phases acetonitrile (A), water 0.1% formic acid (B), and methanol (C). The flow rate was 0.4 mL min^−1^. Gradient started with 2: 98:0% A:B:C isocratic for 5 min, then 8:90:2% A:B:C at 25 min, 30:60:10% A:B:C at 40 min, 70:20:10% A:B:C at 45 min isocratic for three minutes, 2:98:0% A:B:C at 49 min with five minutes for equilibration. The MS parameters were in negative mode for polyphenols and in positive mode for anthocyanin identification. The parameters in negative mode were: capillary voltage 85 volt, RF loading 100%; nebulizer gas pressure, 25 psi; drying gas pressure, 25 psi; drying gas temperature, 265 °C; needle voltage, ±5000 V; spray shield voltage, 600 V. Mass spectra were acquired in negative mode in the spectral range 50–1500 Da. The parameters in positive mode were: capillary voltage 85 volt, RF loading 85%; nebulizer gas pressure, 25 psi; drying gas pressure, 25 psi; drying gas temperature, 265 °C; needle voltage, ±4000 V; spray shield voltage, 600 V. Mass spectra were acquired in negative mode in the spectral range 50–1800 Da. The data obtained were used to tentatively assign an identification to the secondary metabolite compound on the base of the literature references in the MS^n^ pathway.

The amounts of anthocyanins were quantified with the DAD detector. Cyanidin was used as the standard compound and stock solutions were prepared at a concentration of 60 ug ml^−1^ in methanol 1% HCl. Then, the dilutions were prepared in methanol in the range between 60 and 6 ug ml^−1^ and the calibration curve was as follows: y = 15,766x + 28,663 (R^2^ = 0.9999). Peaks with UV spectra with a maximum at 540 nm, ascribable to anthocyanin structure, were selected and the area was obtained at 540 nm. Quantitative data of the various samples were reported as the sum of all of the different quantified constituents. For qualitative purpose, the mass fragmentation pathway was used for tentative identification and each peak was assigned to anthocyanin derivatives.

Polyphenols were also quantified with the DAD detector. Catechin and rutin were used as the standard compound and stock solutions were prepared at a concentration of 60 ug ml^−1^ in methanol. Then, the dilutions were prepared in methanol in the range between 60 and 6 ug ml^−1^, calibration curves were obtained and were y = 69.672 x−515.5 (R^2^ = 0.9984) and y = 11.527 x−35.8 (R^2^ = 0.9884), respectively, for catechin and rutin, respectively.

For qualitative purposes, the mass fragmentation pathway was used for tentative identification and peaks were assigned as catechin and flavonoid proanthocyanin derivatives. Peaks with UV spectra with the maximum at 280 nm ascribable to the catechin structure and identified by the *m/z* value as catechin derivatives, were selected and area was obtained at 280 nm. Peaks with UV spectra with a maximum at 355 nm ascribable to the flavonoid structure and identified by the *m/z* value as a flavonoid, were selected and the area was obtained at 355 nm.

### 4.5. Amino Acid Analysis

To assess the amount of amino acids, we based our determination on our previous method [33] with small changes. The Agilent 1260 chromatograph (Santa Clara, CA, USA) equipped with a Varian MS-320 triple quadrupole mass spectrometer with an ESI source was used. The LC separation was obtained on an Agilent Z-Hilic 3.0 × 10 mm 4.6 micron), as mobile phases acetonitrile (A), water 0.1% formic acid (B) were used. The flow rate was 0.4 mL min^−1^. The gradient started with 2: 98:0% A:B:C isocratic for 5 min, then 8:90:2% A:B:C at 25 min, 30:60:10% A:B:C at 40 min, 70:20:10% A:B:C at 45 min isocratic for three minutes, and 2:98:0% A:B:C at 49 min with five minutes for equilibration. Standard solutions were prepared, weighing the exact amount of each amino acid in diluted formic acid solution (5%), diluting then in water to obtain concentrations of 10, 5, 2, and 1 µg mL^−1^. The solutions were centrifuged and used for analysis. For each amino acid, specific transitions were optimized by the direct infusion of standard solution at 1 ug/mL and the selected transitions used for the quantification are reported in the Appendix A (Appendix A).

### 4.6. Sugar Analysis and Tartaric Acid Quantification

Sugars and tartaric acid were quantified using an Agilent 1100 chromatograph (Santa Clara, CA, USA) equipped with an evaporative light scattering detector (ELSD) (Sedex, Sedere, Olivet, France). The LC separation was obtained on a Hi-plex H column (300 × 7.7 mm), as previously reported [34], as mobile phases of water 0.002% trichloride acetic acid were used. The flow rate was 0.6 mL min^−1^ and the column temperature was set at 75 °C. The ELSD detector parameter was 70 °C, 2.5 bar, and gain 7. Glucose, fructose, and tartaric acid were used as the reference compounds and standard solutions were prepared by exactly weighing 30 mg of each compound in 10 mL of water, and then diluted to obtain a calibration curve.

### 4.7. Multivariate Analysis

The targeted metabolomic approach was used and data modeling was performed by applying a multivariate technique based on projection. Principal component analysis (PCA) was used for exploratory data analysis and to highlight the differences in the composition between the various grouped samples. A matrix was created by merging data obtained by LC-DAD-ESI-MS for the phenolic constituents and for amino acids as well as the LC-ELDS for sugars and organic acids. The amount of each compound was used, and the data were Pareto scaled. Fifty different compounds were used as descriptors of the berry samples. The data matrix was imported in SIMCA 12 (Umetrics, Umeå, Sweden) and used to perform PCA, PLS-DA and OPLS-DA analysis, as previously reported [20].

### 4.8. Statistical Analysis

Quantitative results obtained from the measurements of amino acids, flavonoids, anthocyanins, and PAC were subjected to a one-way analysis of variance (ANOVA) and a comparison between the treated and control group was performed by the *t*-test. All the experiments were performed in triplicate. The targeted metabolomics analysis was obtained by the combination of the results of the previous analytes and quantitative measurements were transformed in a matrix that was imported into Simca 12 software (Umetrix, Umeå, Sweden). The unsupervised method, namely PCA, and supervised OPLS-DA and PLS-DA were performed using centering and Pareto scaling. The statistical analyses were validated by performing the permutation test (150 permutations), CV-ANOVA (*p* < 0.05), and *t*-test (*p* < 0.05) for the considered metabolites.

## 5. Conclusions

In this work, the target metabolomic approach was used to compare the possible changes induced by Si-treatment in the berry composition of four different cultivars. PAC, flavonoids, anthocyanins, sugars, and amino acids were quantified by different analytical approaches, yielding a detailed chemical characterization of the berries’ metabolites. Each cultivar showed peculiar variations in specific metabolites. A different behavior was described for red and white grapes. Namely, the anthocyanin contents increased in the red grapes while proline and PAC were enhanced only in white grapes. The results indicate that some modifications related to the Si treatment occurred in each of the four grape cultivars, but no common trend in metabolite changes could be observed. The overall results showed the usefulness of the metabolomic approach in the study of the effects of food plant treatments, having detailed chemical information on the final product composition. Further studies are needed to deeply understand the role of silicon fertilization in other environmental conditions, during biotic and abiotic stress, and with different grape cultivars, also considering other classes of the berry constituents.

## Figures and Tables

**Figure 1 plants-11-02998-f001:**
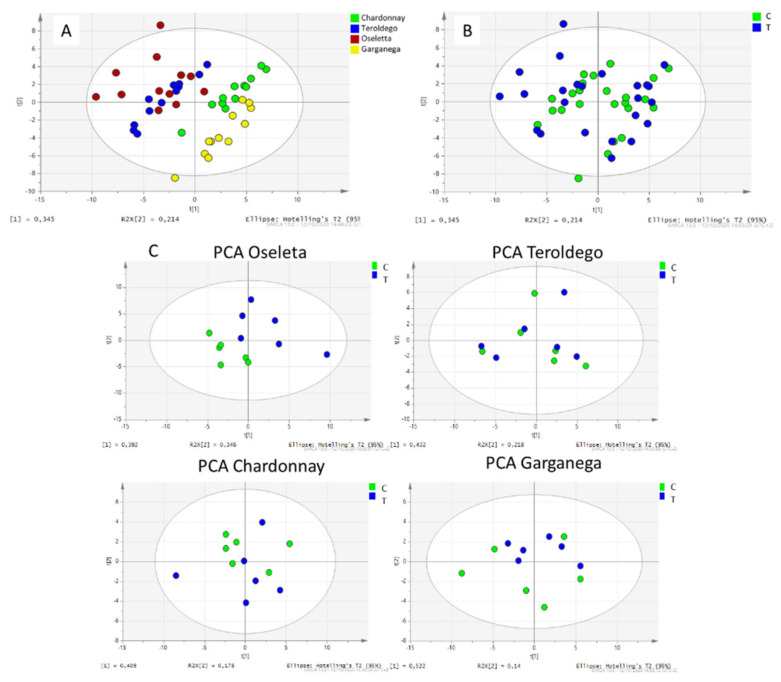
(**A**) PCA loading plot of the control and treated berries of Oseleta (red) Teroldego (blue) Chardonnay (green), and Garganega (yellow). (**B**) PCA loading plot of all the samples. Blue dots represent silicontreated samples, green dots represent control samples. (**C**) PCA loading plot of the treated and control sample of Oseleta (above-left), Teroldego (above-right), Chardonnay (below-left) and Garganega (below-right). Blue dots represent the silicon-treated samples, green dots represent the control samples.

**Figure 2 plants-11-02998-f002:**
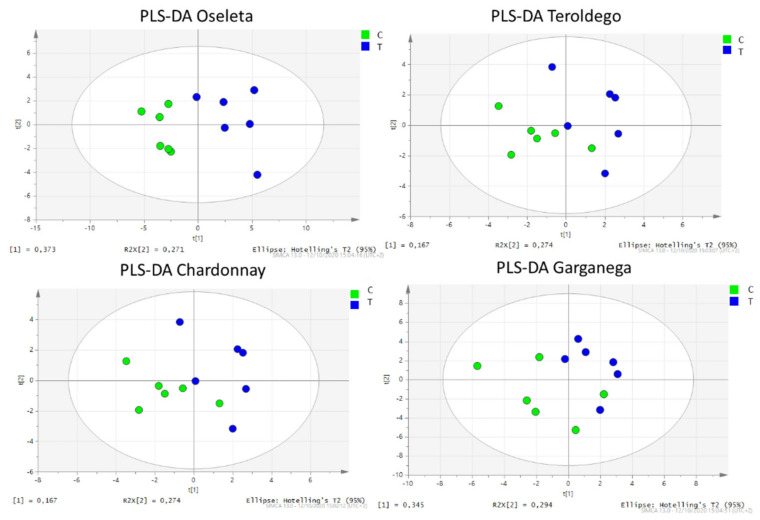
PLS-DA loading plot of the treated and control samples of Oseleta (above-left), Teroldego (above-right), Chardonnay (below-left), and Garganega (below-right). Blue dots represent silicon-treated samples, green dots represent control samples.

**Figure 3 plants-11-02998-f003:**
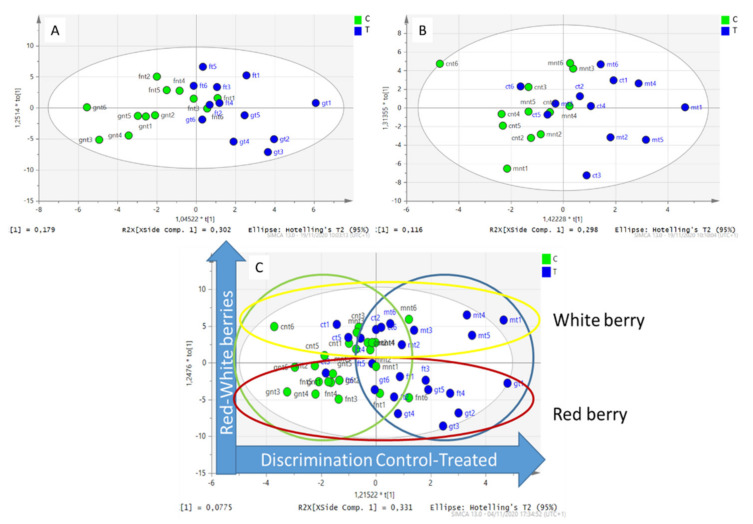
(**A**): OPLS-DA loading plot of the red grape samples (Oseleta, Teroldego). Blue dots represent the silicon-treated samples, green dots represent the control samples. (**B**) OPLS-DA loading plot of the white grape samples (Chardonnay, Garganega). Blue dots represent the silicon-treated samples, green dots represent the control samples, (**C**) OPLS-DA Loading plot of all grape samples. Blue dots represent the silicon-treated samples, green dots represent the control samples. Green and blues circles highlight the clusterization between the treated and control samples, yellow and red circles highlight the clusterization between the white and red berries.

**Table 1 plants-11-02998-t001:** Identified secondary metabolites and their quantification in the berries of the four grape cultivars. RT: retention time (min), ion, and fragments (*m/z*). Data are expressed in mg g^−1^ of fresh material.

RT	Ion	Fragments	Identification	Oseleta	Teroldego	Garganega	Chardonnay
	[M + H]^+^		**Anthocyanidins**				
25.9	465	303	delphinidin-3-O-glucoside	0.10 ± 0.01	0.114 ± 0.01	-	-
29.8	448	287	cyanidin-3-O-glucoside	0.03 ± 0.010	0.233 ± 0.001	-	-
31.5	479	317	petunidin-3-O-glucoside	0.083 ± 0.005	0.118 ± 0.005	-	-
32.9	463	301	peonidin-3-O-glucoside	0.066 ± 0.001	0.454 ± 0.001	-	-
33.4	493	331	malvidin-3-O-glucoside	0.284 ± 0.003	0.315 ± 0.002	-	-
34.4	507	303	delphinidin-3-O-(6″-acetyl-glucoside)	0.018 ± 0.002	0.003 ± 0.002	-	-
35.5	491	287	cyanidin-3-O-(6″-acetyl-glucoside)	0.007 ± 0.005	0.002 ± 0.001	-	-
35.9	521	317	petunidin-3-O-(6″-acetyl-glucoside)	0.021 ± 0.003	0.001 ± 0.001	-	-
37	505	301	peonidin-3-O-(6″-acetyl-glucoside)	0.118 ± 0.001	0.009 ± 0.001	-	-
37.3	611	303	delphinidin-3-O-(6-*p*-coumaroyl)glucoside				
37.7	535	331	malvidin-3-O-(6″-acetyl-glucoside)				
38	655	331	malvidin-3-O-(6″-caffeoyl-glucoside)	0.011 ± 0.005	0.009 ± 0.001	-	-
38.1	595	287	cyanidin-3-O-(6-*p*-coumaroyl)glucoside	0.017 ± 0.003	0.005 ± 0.001	-	-
38.3	625	317	petunidin-3-O-(6-*p*-coumaroyl)glucoside				
39	639	331	malvidin-3-O-(6-*p*-coumaroyl)glucoside	0.097 ± 0.001	0.043 ± 0.002	-	-
39.1	609	301	peonidin-3-O-(6-*p*-coumaroyl)glucoside				
			total amount	**0.852**	**1.306**		
	[M - H]^-^		**Flavonoids**				
34.9	463	301	quercetin-7-O-glucoside	-	-	0.008 ± 0.001	0.004 ± 0.001
35.2	477	301	quercetin-O-glucuronide	-	-	0.01 ± 0.001	0.005 ± 0.001
35.3	463	301	quercetin-3-O-glucoside	-	-	0.033 ± 0.001	0.002 ± 0.001
37	477	285	kaempferol-O-hexoside	-	-	0.006 ± 0.001	-
			total amount			**0.057**	**0.011**
	[M - H]^-^		**PAC derivatives**				
16.2	577	425 407	pac dimer isomer 1	0.01 ± 0.001	0.013 ± 0.001	0.019 ± 0.001	0.027 ± 0.001
17.1	289	245 205	catechin	0.02 ± 0.005	0.043 ± 0.005	0.013 ± 0.002	0.054 ± 0.005
19.8	865	695 577	pac trimer isomer 1	0.029 ± 0.002	0.043 ± 0.002	0.027 ± 0.005	0.021 ± 0.001
21.4	443	-	unknown	-	-	0.002 ± 0.001	-
23.2	577	425 407	pac dimer isomer 2	0.020 ± 0.006	0.025 ± 0.002	0.013 ± 0.001	0.010 ± 0.001
26.5	289	245 205	epicatechin	0.020 ± 0.003	0.040 ± 0.008	0.020 ± 0.006	0.059 ± 0.003
27	729	577 407	procyanidin dimer monogallate	0.014 ± 0.001	0.005 ± 0.001	0.003 ± 0.001	-
31.2	865	695 577	pac trimer isomer 2	0.003 ± 0.001	0.023 ± 0.002	0.019 ± 0.001	0.005 ± 0.001
32	1153	865 575	pac tetramer	-	0.007 ± 0.001	0.004 ± 0.001	-
33	729	577 407	procyanidin dimer monogallate	-	-	-	0.001 ± 0.001
33.7	441		catechin gallate	-	0.017 ± 0.001	0.018 ± 0.001	-
35.3	865	695 577	pac trimer isomer 3	-	0.033 ± 0.002	-	-
			total amount	**0.116**	**0.249**	**0.138**	**0.177**

**Table 2 plants-11-02998-t002:** Total PAC, anthocyanins, flavonoids, amino acids, tartaric acid, glucose, and fructose content in the four considered cultivars. Columns represent the control group C and treated group T. Total PAC, anthocyanins, flavonoids, tartaric acid, glucose, and fructose are expressed in mg g^−1^ fresh weight, amino acids are expressed in mg kg^−1^ fresh weight. * Statistically different (*p* < 0.05).

	Oseleta	Teroldego	Chardonnay	Garganega
Tot PAC C	0.09 ± 0.03	0.26 ± 0.07	0.22 ± 0.05	0.13 ± 0.06
Tot PAC T	0.14 ± 0.04	0.24 ± 0.08	0.17 ± 0.05	0.15 ± 0.04
anthocyanin C	0.48 ± 0.23 *	1.31 ± 0.19	-	-
anthocyanin T	1.25 ± 0.33 *	1.34 ± 0.18	-	-
flavonoid C	-	-	0.010 ± 0.004	0.055 ± 0.018
flavonoid T	-	-	0.012 ± 0.006	0.060 ± 0.015
amino acids C	467 ± 51	398 ± 51 *	443 ± 52	394 ± 98
amino acids T	444 ± 48	590 ± 42 *	467 ± 90	429 ± 87
tartaric acid C	0.93 ± 0.4 *	2.5 ± 0.5	0.93 ± 0.35 *	1.09 ± 0.16
tartaric acid T	1.21 ± 0.2 *	2.6 ± 0.3	0.66 ± 0.13 *	1.09 ± 0.19
glucose C	117.7 ± 9.6	103 ± 4.7	115.4 ± 8.0	91.5 ± 14.9
glucose T	121.2 ± 4.4	98.4 ± 5.4	111.7 ± 10.5	84.5 ± 9.2
fructose C	98.6 ± 7.4	95.4 ± 3.7 *	98.7 ± 7.0	75.7 ± 9.6
fructose T	103.5 ± 4.7	88.4 ± 4.4 *	94.5 ± 7.1	71.6 ± 8.8

**Table 3 plants-11-02998-t003:** Descriptive metabolites of Oseleta, Teroldego, Garganega, and Chardonnay berries expressed as % variations of T vs. C samples. * indicates *p* < 0.05.

Samples	Metabolites	T vs. C (%)
Oseleta	valine	−54.7 *
	arginine	−81.3 *
	phenyl alanine	−31.2 *
	asparagine	−16.3 *
	malvidin-3-O-glucoside	+119.9 *
	peonidine-3-O-glucoside	+318.3 *
	petunidin-3-O-(6″-acetyl-glucoside)	+176.6 *
	cyanidin-3-O-(6-*p*-coumaroyl)glucoside/petunidin-3-O-(6-*p*-coumaroyl)glucoside	+141.2 *
Teroldego	fructose	−7.6 *
	glucose	−5.1
	procyanidin trimer isomer 1	−16.9
	cyanidin-3-O-(6″-acetyl-glucoside)	−34.0 *
	asparagine	+29.4
	procyanidin trimer isomer 2	+28.6
	malvidin-3-O-(6-*p*-coumaroyl)glucoside/peonidin-3-O-(6-*p*-coumaroyl)glucoside	+15
	delphinidin-3-O-(6″-acetyl-glucoside)	+11.1
Garganega	kaemferol O hexoside	−13.8
	glucose	−7.6
	fructose	−5.4
	arginine	−9
	proline	+105.1 *
	procyanidin trimer isomer 2	+26.6
	procyanidin tetramer	+132.8
	catechin gallate	+62.1
Chardonnay	procyanidin trimer isomer 2	−27.7
	catechin	−32
	tartaric acid	−28.8 *
	epicatechin	−26
	histidine	+3.02
	tyrosine	+1.37
	serine	+1.83
	leucine	+0.67

## Data Availability

Not applicable.

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
