# Peer review of "Foliar Application of Silicon in Vitis vinifera: Targeted Metabolomics Analysis as a Tool to Investigate the Chemical Variations in Berries of Four Grapevine Cultivars"

_plants, 2022, doi:10.3390/plants11212998_

Round 1
Reviewer 1 Report
Journal: Plants (ISSN 2223-7747)
Manuscript ID: plants-1969184
Manuscript Title: "Foliar application of silicon in Vitis vinifera: targeted metabolomics analysis as tool to investigate the chemical variations in berries of four grapevine cultivars"
Reviewer comments:
After reviewing the entire manuscript, it was found that it was written in sound scientific language and that its results are good, but it is very long and needs to be abbreviated without repetition, and that the tables and figures are clear, also that the discussion part needs more depth and linking some sentences with each other to reach clear scientific information. The materials and methods section needs to be supported by relevant modern references.
The conclusion is that the manuscript can be accepted for publication after making the required modifications and after the approval of the academic editor.
My comments on the manuscript are in the attached file.

Author Response
Please see in the attachment

Reviewer 2 Report
Comments to the authors:
The paper entitled “Foliar Application of Silicon in Vitis vinifera: Targeted Metabolomics Analysis as Tool to Investigate the Chemical Variations in Berries of Four Grapevine Cultivars.” is an original research article. The paper needs to be revised carefully. The article has novelty and the method is modern.
The topic selection and research methods of this paper are relatively good. The author has also done a lot of research work, the research content is relatively rich, and the relevant data are also relatively detailed. However, the author lacks sufficient logic in the background analysis, problem generation, problem analysis, and discussion of the results, and some sentences have grammatical errors, which lead to a disconnect between the theme and content of this paper.
1. The abstract should highlight the key points and strengthen the logic between the research background, content, results, analysis, and discussion, rather than listing all the results. Moreover, the highlights and key points of this work should be highlighted in the abstract. Also, the abstract need to be rearranged see more articles on how to write an abstract.
2. The keywords need to be different from the words used in the title of the manuscript.
3. The introduction needs to focus on the literature related to the topic. The point lack in this section is connectivity between the paragraphs and sentences. This section needs detailed attention.
4. The methodology overall is not in an organized form, need to improve the presentation of this section.
5. The results and discussion need to be separate, and the comparative discussion needs to be stronger. Needs more evidence from the previous literature to compare with the current results obtained.
6. The sentences in the text should be scientific and standardized and should be carefully checked in the overall manuscript (for example, please refer to other literature for the method section).
Author Response
Please see in the attachment
